# Examining the Association between Neighbourhood Socioeconomic Disadvantage and Type 2 Diabetes Comorbidity in Serious Mental Illness

**DOI:** 10.3390/ijerph16203905

**Published:** 2019-10-15

**Authors:** Ramya Walsan, Darren J Mayne, Xiaoqi Feng, Nagesh Pai, Andrew Bonney

**Affiliations:** 1School of Medicine, Faculty of Science, Medicine and Health, University of Wollongong, Wollongong 2522, Australia; darren.mayne@health.nsw.gov.au (D.J.M.); nagesh@uow.edu.au (N.P.); abonney@uow.edu.au (A.B.); 2Illawarra Health and Medical Research Institute, Wollongong 2522, Australia; xfeng@uow.edu.au; 3Illawarra Shoalhaven Local Health District, Public Health Unit, Warrawong 2502, Australia; 4The University of Sydney, School of Public Health, Sydney 2006, Australia; 5Population Wellbeing and Environment Research Lab (PowerLab), School of Health and Society, Faculty of Social Sciences, University of Wollongong, Wollongong 2522, Australia; 6School of Public Health and Community Medicine, University of New South Wales, Kennington 2031, Australia; 7Mental Health Services, Illawarra Shoalhaven Local Health District, Wollongong Hospital, Wollongong 2500, Australia

**Keywords:** neighbourhood disadvantage, serious mental illness, type 2 diabetes, comorbidity

## Abstract

This study examined the association between neighbourhood socioeconomic disadvantage and serious mental illness (SMI)–type 2 diabetes (T2D) comorbidity in an Australian population using routinely collected clinical data. We hypothesised that neighbourhood socioeconomic disadvantage is positively associated with T2D comorbidity in SMI. The analysis considered 3816 individuals with an SMI living in the Illawarra and Shoalhaven regions of NSW, Australia, between 2010 and 2017. Multilevel logistic regression models accounting for suburb (neighbourhood) level clustering were used to assess the association between neighbourhood disadvantage and SMI -T2D comorbidity. Models were adjusted for age, sex, and country of birth. Compared with the most advantaged neighbourhoods, residents in the most disadvantaged neighbourhoods had 3.2 times greater odds of having SMI–T2D comorbidity even after controlling for confounding factors (OR 3.20, 95% CI 1.42–7.20). The analysis also revealed significant geographic variation in the distribution of SMI -T2D comorbidity in our sample (Median Odds Ratio = 1.35) Neighbourhood socioeconomic disadvantage accounted for approximately 17.3% of this geographic variation. These findings indicate a potentially important role for geographically targeted initiatives designed to enhance prevention and management of SMI–T2D comorbidity in disadvantaged communities.

## 1. Introduction 

Serious mental illness (SMI) is a term used to refer to severe and persistent forms of mental disorders such as schizophrenia, bipolar disorder or major depression [1]. Individuals with SMI have 2 to 4 times increased risk of developing type 2 diabetes (T2D) compared with the general population which translates into a reduction of 15–20 years in their life expectancies [2,3,4]. A comorbid T2D diagnosis is also associated with other adverse consequences such as increased hospitalisations, greater number of emergency department visits, non-adherence to treatments, higher healthcare utilisation costs, higher risk of cognitive deficit, poor clinical outcomes and decreased quality of life for the mentally ill. [2,5,6,7,8,9,10,11]. 

People with SMI are more likely to live in disadvantaged neighbourhoods [12,13] and the environment in these neighbourhoods may compound the experiences of psychosocial stress or promote engagement in adverse health behaviours (e.g. unhealthy eating and physical inactivity) and weight gain, all of which contribute to T2D risk [12,14,15]. A number of studies have found that the prevalence of SMI and T2D are both separately higher in more socioeconomically disadvantaged neighbourhoods [13,16,17,18,19]. However, research to date has not adequately examined the association between area-level disadvantage and SMI–T2D comorbidity. A recent systematic review [20] examining this relationship identified only a single study demonstrating a tentative association between the neighbourhood level disadvantage and T2D comorbidity in mental illness [21]. The aforementioned study, however, focused entirely on major depression and did not consider other forms of SMI such as schizophrenia or bipolar disorder. Hence additional research on the association between neighbourhood disadvantage and SMI–T2D comorbidity is warranted, given the paucity of evidence available and the plausibility of an association. We have recently reported significant geographic variations in the distribution of SMI–T2D comorbidity suggesting the need to explore the role of neighbourhood level disadvantage in explaining this variation [22]. 

Establishing strong evidence of the relationship between neighbourhood disadvantage and SMI –T2D comorbidity is an important step in advancing our understanding of the T2D comorbidity in SMI and the possible associations with neighbourhood environments might have with this comorbidity. Moreover, population-based prevention strategies that shift the risk distribution of entire population in a favourable direction are considered more effective and sustainable than the individual-based approaches in reducing the disease burden [23]. Understanding these associations may also be useful for health policymakers to develop integrated interventions and to provide greater diversity of care needed to optimally manage the complex needs associated with comorbidity.

The aim of this study was to investigate the association between neighbourhood socioeconomic disadvantage and SMI–T2D comorbidity in an Australian population using routinely collected clinical data. We hypothesised that greater socioeconomic disadvantage would be associated with increased T2D comorbidity in SMI. A further objective was to determine how much variance of SMI-T2D comorbidity between neighbourhoods was attributable to neighbourhood socioeconomic disadvantage. 

## 2. Materials and Methods 

### 2.1. Study Design and Sample

We used a cross-sectional, multilevel study design to examine the association between neighbourhood socioeconomic disadvantage and SMI-T2D comorbidity. The study area comprised the Illawarra and Shoalhaven regions of NSW, Australia, which had an estimated resident population of 368,604 people at the time of the 2011 Australian Census of Population and Housing [24]. The region has a mix of rural and urban influences and is comprised of the local government areas of Kiama, Shellharbour, Shoalhaven, and Wollongong. The socio-economic profile of the study area as described by region’s socio-economic index scores are comparable to that of NSW and Australian average [25,26]. The data analysed in this study covered the period 01 January 2010 to 31 December 2017 and were retrieved from Illawarra Health Information Platform (IHIP). The IHIP is a research partnership established between Illawarra Shoalhaven Local Health District (ISLHD) and University of Wollongong for the purpose of providing ISLHD health service data to clinicians and researchers. The analysis was undertaken at the state suburb level (SSCs), which was the smallest geographic unit at which the health service data were available. State suburbs are the Australian Bureau of Statistics (ABS) approximation of suburbs gazetted by the Geographical Names Board of NSW [27]. The Illawarra-Shoalhaven region comprised of 167 suburbs with an average land area of 36.56 km^2^ and 2207 residents each in 2011 [24]. 

This study was approved by the University of Wollongong and Illawarra Shoalhaven Local Health District Human Research Ethics Committee (protocol number 2017/428).

### 2.2. Measures

Data extraction was carried out using International Classification of Diseases (ICD) version 10 codes and was restricted to adults 18 years and over. We defined SMI as having a primary or secondary diagnosis of schizophrenia (F20), other non-affective psychosis (F22–F29), bipolar disorder (F30, F31), major depression (F32, F33) or other affective disorders (F34, F39) in the inpatient records of ISLHD. Diabetes comorbidity, the outcome of interest, was defined as having a T2D diagnosis (E11) in people with SMI and was extracted as either present or absent along with each of the SMI records. The analytical sample was formed by excluding individuals residing outside the Illawarra and Shoalhaven regions (*n* = 50) and individuals with no suburb (*n* = 283) or country of birth information (*n* = 8). The final SMI sample consisted of 3816 individuals of whom 463 (12.09 %) had T2D comorbidity. 

Neighbourhood socioeconomic disadvantage was operationalised for suburbs using the Index of Relative socioeconomic disadvantage (IRSD) from the 2011 Socioeconomic Indexes for Area Census product [26]. An IRSD score reflects the aggregate level of socioeconomic disadvantage measured on the basis of 17 variables including education, income, occupation, unemployment, housing type, overcrowding, and English proficiency. For this study, IRSD scores for Illawarra and Shoalhaven suburbs were divided into quintiles of neighbourhood disadvantage with Quintile one (Q1) denoting the 20% most disadvantaged suburbs in Illawarra-Shoalhaven and Quintile five (Q5) the least disadvantaged 20%. Global Moran’s I revealed a significant spatial dependence for neighbourhood socioeconomic disadvantage quintiles (Moran’s I = 0.443673, *p* < 0.0001) indicating that suburbs with similar relative neighbourhood disadvantage are clustered geographically [28]. Quintiles were then assigned to individuals based on their suburb of residence at their most recent admission before 31 December 2017. 

Individual-level variables included in the analysis were sex, age at most recent admission and the country of birth. Age was categorised into three age groups (18–44, 45–65, 65+) and sex were grouped as male or female. Country of birth data was aggregated based on the *Standard Australian Classification of Countries* produced by the Australian Bureau of Statistics [29]. 

### 2.3. Statistical Analysis

Multilevel logistic regression models accounting for suburb level clustering were used to assess the association between neighbourhood disadvantage and SMI–T2D comorbidity. The data structure consisted of two levels with individuals (level 1) nested within suburbs (level 2). A series of models were fit as follows: model 1 included only suburb level random effect, model 2 added individual level factors (age, gender, country of birth) to model 1, and model 3 added neighbourhood level IRSD quintiles to model 2. Interactions between individual variables and neighbourhood disadvantage were also considered in modelling to investigate any cross-level effect modification of the association by individual-level factors. Models were estimated using the maximum likelihood method with Laplace approximation [30]. Intra class correlation (ICC) and Median Odds ratios (MOR) were calculated for each model to assess how much of the variance in comorbidity could potentially be attributed to neighbourhoods [31,32]. ICC informs us regarding the variance between areas [31]. The MOR is interpreted as the increased risk in comorbidity when an individual moves to a suburb of higher disadvantage [33]. MOR closer to 1 implies little variation between areas whereas larger MOR values indicate considerable variation between areas [33]. We also reported proportional change in variance (PCV) to show how much of the residual variance was explained by the additional explanatory variables in each of the models. ICC, MOR, and PCV were derived from model outputs following the methods specified by Merlo et al and Austin et al [31,32]. Likelihood ratio tests were used to determine the goodness of fit of the models. All statistical analysis was completed using R version 3.5 [34]. Statistical significance in this analysis was set at *p* < 0.05.

## 3. Results

The descriptive characteristics of the study population are given in Table 1. SMI-T2D comorbidity was present in 13.3% of females and 11.1% of males with an SMI diagnosis. The age group with highest proportion of comorbidity was 65+ (27.73%). With regards to country of birth, a higher percentage of T2D comorbidity was observed for SMI individuals born in Middle East and North Africa (23.1%), Eastern and Central Europe (23.2%) and Western Europe (21.2%). The SMI-T2D comorbidity prevalence in the most disadvantaged IRSD quintile (Q1) was 13.1% (*n* = 229) and that in the least disadvantaged quintile (Q5) was 5.1% (*n* = 7). 

Table 2 presents the results of the multilevel logistic regression analysis. Model 1 provides the estimate of between area variation in SMI–T2D comorbidity without any explanatory variables. The MOR for model 1 was 1.35, indicating some level of geographic variation in the distribution of SMI-T2D comorbidity in our sample. Moreover, the ICC for model 1 was 0.029, showing that 2.9% of the variance in comorbidity was attributable to between neighbourhood differences. The addition of individual level variables in model 2 accounted for 25.5% of between area variance and addition of IRSD in model 3 accounted for an additional 17.3% and reduced the MOR to 1.25. After inclusion of individual and neighbourhood variables, the ICC decreased from 2.9% to 1.7%. 

Results for individual-level variables in Model 2 indicate that age was significantly associated with SMI-T2D comorbidity. Older individuals with SMI have significantly higher odds of having T2D comorbidity compared with younger individuals. Model 3 showed a significant association between higher levels of neighbourhood disadvantage and diabetes comorbidity in SMI after controlling for age, gender and country of birth. Living in a neighbourhood with the highest socioeconomic disadvantage was associated with 3 times increased odds of having SMI-T2D comorbidity compared with the least disadvantage neighbourhood (OR 3.20, 95% CI 1.42–7.20 for Q1 vs Q5). Including two-way interaction terms in Model 3 indicated no evidence of effect modification of the association between SMI–T2D comorbidity and IRSD by age (χ^2^_LRT_ = 14.16, DF = 8, *p* = 0.077), gender (χ^2^_LRT_ = 1.45, DF = 4, *p* = 0.835) or country of birth (χ^2^_LRT_ = 30.68, DF = 38, *p* = 0.794). 

## 4. Discussion

We found an independent positive association between neighbourhood disadvantage and SMI–T2D comorbidity after controlling for individual age, gender and country of birth. Neighbourhood socioeconomic disadvantage accounted for 17.3% of the between neighbourhood variation in SMI–T2D comorbidity. Among the individual-level factors, age was independently associated with SMI–T2D comorbidity. Individual factors accounted for 25.5% of the between neighbourhood variation. Neither gender nor country of birth were associated with SMI-T2D comorbidity. Lower neighbourhood variance in SMI–T2D comorbidity (ICC = 0.029) reported in our study does not preclude important neighbourhood level effects [35]. Misspecification of neighbourhoods, smaller group sizes and even omission of a relevant level 1 variable can all cause under estimation of neighbourhood variance [36]. Low Intra class correlation (ICC) can coexist with important neighbourhood level fixed effects and several of these examples are available in public health where risk factors explain very little neighbourhood variance but are important predictors of health outcomes [36]. Additionally, Geoffrey Rose had pointed out that even small neighbourhood effects, when aggregated at population scales, can have a massive impact [23]. Ours appears to be one of the first studies to explore the association between area level disadvantage and SMI–T2D comorbidity. The only other study addressing this research question investigated major depression only and reported a positive but non-significant association between area level disadvantage and SMI–T2D comorbidity [21]. Our findings are, however, consistent with prior studies, which show significant neighbourhood level socioeconomic inequalities in the distribution of SMI [13,17,37] and T2D [18,19,38,39] as independent conditions. In their systematic review, Mair et al identified 45 studies, of which 37 reported significant associations between neighbourhood characteristics and depression [40]. Similarly, the significant associations between neighbourhood environments and T2D risk were revealed in another systematic review by Dendup et al. [41]. The findings of a positive significant association between SMI–T2D comorbidity and age and a non-significant association between SMI–T2D comorbidity and gender are consistent with previous reports in the literature [3,42,43]. 

The results of this study have policy implications for planning interventions and resourcing public health services. Our results indicate that efforts to reduce diabetic comorbidity in serious mental illness might benefit by focussing on individuals with SMI living in high deprivation neighbourhoods. These results also have future research implications. Understanding why neighbourhood level disadvantage is associated with comorbidity is an important next step in addressing these inequities and in developing sustainable interventions and long-term solutions. There are several plausible explanations for increased SMI–T2D comorbidity in more disadvantaged neighbourhoods, over and above individual level factors. Neighbourhood level features, such as green spaces, access to health care services, availability of fast food restaurants and area level crime may be differentially present in advantaged and disadvantaged neighbourhoods [44]. These may in turn act as a stimulus for chronic stress or adverse health behaviours such as unhealthy eating, lack of physical activity and obesity, which have been shown to be associated with increased T2D risk [12,14,15]. Further exploration of the mediating or confounding roles played by these contextual variables may improve our understanding of SMI–T2D comorbidity and the casual pathways linking them with the neighbourhood environments. 

There are some limitations to our study. First, the cross-sectional study design does not allow us to draw cause-effect conclusions. Second, we used data sourced only from inpatient mental health records and did not consider outpatient and private practice records. However, the Australian National Surveys of Psychosis indicates that 45.6–62.9% of people with SMI reported ≥1 hospital admission for any reason in the previous 12 months [45], which should have provided a reasonable coverage given our eight-year data collection period. In addition, we acknowledge the potential for temporal misalignment as 2011 relative disadvantage index scores were used in this analysis. Nonetheless a weighted Kappa analysis between 2011 and 2016 disadvantage quintiles revealed a good agreement between the two (k = 0.796) indicating that the deprivation scores have remained relatively similar during these periods. Individual socioeconomic status, ethnicity, age at diagnosis and number of hospital admissions, were not included in this analysis due to the lack of data availability. This may have resulted in the overestimation of neighbourhood level effects. Finally, we also acknowledge the potential for reverse causation as SMI and T2D share a bidirectional association. 

## 5. Conclusions

Our results indicate that the people with SMI living in the most disadvantaged neighbourhoods are more likely than their counterparts in the least disadvantaged neighbourhoods to report SMI-T2D comorbidity. These findings highlight the need to consider public health prevention strategies at both individual and neighbourhood level in order to reduce the public health burden imposed by comorbidity. The current study makes a significant contribution to the scant research literature available in this area of public health. Future research is needed to extend these findings and to consider how various neighbourhood contextual features may mediate the effect of neighbourhood socioeconomic disadvantage on SMI–T2D comorbidity. 

## Figures and Tables

**Table 1 ijerph-16-03905-t001:** Characteristics of study population Variables.

Variables	Individuals with SMI *n* = 3816	Individuals with SMI–T2D Comorbidity *n* = 463	% of Individuals with SMI who Also Have Comorbidity (95% Cl)
Individual variables			
Gender			
Female	1848 (48%)	245 (53%)	13.3 (11.8–14.9)
Male	1968 (52%)	218 (47%)	11.1 (9.7–12.5)
Age, years (Mean (SD))			
Age, years	43.6 (18.5)	58.8 (15.7)	
18–44	1961 (51%)	92 (20%)	4.7 (03.8–05.7)
45–65	1213 (32%)	193 (42%)	15.9 (13.9–18.0)
65+	642 (17%)	178 (38%)	27.7 (24.3–31.2)
Country of birth			
Australia	3104 (81%)	339 (73%)	10.9 (9.9–12.1)
Oceania excluding Australia	74 (2%)	12 (3%)	16.2 (9.5–26.2)
UK & Ireland	212 (6%)	35 (8%)	16.5 (12.1–22.1)
Western Europe	137 (4%)	29 (6%)	21.2 (15.2–28.8)
Eastern and Central Europe	125 (3%)	29 (6%)	23.2 (16.7–31.3)
North East Asia	17 (0%)	0 (0%)	0.0 (0–18.4)
South East Asia	51 (1%)	6 (1%)	11.8 (5.5–23.4)
Central and South Asia	16 (0%)	3 (1%)	18.8 (6.6–43.0)
Middle East and North Africa	39 (1%)	9 (2%)	23.1 (12.7–38.3)
Sub-Saharan Africa	20 (1%)	0 (0%)	0.0 (0–16.1)
Americas	21 (1%)	1 (0%)	4.8 (0.9–22.7)
Neighbourhood level variables			
IRSD as quintiles			
Q1 (Highest)	1752 (46 %)	229 (49%)	13.1 (11.6–14.7)
Q2	943 (25 %)	120 (26%)	12.7 (10.7–14.9)
Q3	620 (16 %)	75 (16%)	12.1 (9.8–14.9)
Q4	362 (10 %)	34 (7%)	9.4 (6.8–12.8)
Q5 (Lowest)	139 (4 %)	7 (2%)	5.1 (2.5–10.0)

IRSD = Index of Relative Socioeconomic Disadvantage.

**Table 2 ijerph-16-03905-t002:** The association between neighbourhood socioeconomic disadvantage and serious mental illness (SMI)–type 2 diabetes (T2D) comorbidity using multilevel analysis (Illawarra – Shoalhaven, 2010–2017).

Variable	Model 1	Model 2	Model 3
	OR (95% Cl)	OR (95% Cl)	OR (95% Cl)
Individual variables			
Gender		*p* = 0.658	*p* = 0.687
Female		1.00	1.00
Male		0.95 (0.78–1.17)	0.96 (0.78–1.17)
Age		*p* < 0.05	*p* < 0.05
18–44		1.00	
45–65		3.79 (2.91–4.93)	3.78 (2.90–4.92)
65+		7.68 (5.77–10.23)	7.82 (5.87–10.42)
Country of birth		*p* = 0.137	*p* = 0.149
Australia		1.00	1.00
Oceania excluding Australia		1.57 (0.81–3.03)	1.53 (0.79–2.97)
UK & Ireland		0.84 (0.57–1.26)	0.88 (0.59–1.31)
Western Europe		0.99 (0.63–1.54)	0.97 (0.62–1.52)
Eastern and Central Europe		1.30 (0.82–2.05)	1.30 (0.82–2.06)
South East Asia		1.30 (0.53–3.19)	1.30 (0.52–3.19)
Central and South Asia		2.03 (0.53–7.82)	2.13 (0.56–8.10)
Middle East and North Africa		1.84 (0.83–4.09)	1.87 (0.84–4.16)
Americas		0.42 (0.06–3.25)	0.41 (0.05–3.15)
Neighbourhood Variable			
IRSD quintiles			*p* <0.05
Q5 (Least disadvantaged)			1.00
Q4			1.87 (0.77–4.53)
Q3			2.67 (1.14–6.15)
Q2			2.92 (1.28–6.67)
Q1 (Most disadvantaged)			3.20 (1.42–7.20)
Variance of random effects			
T^2^	0.098	0.073	0.056
PCV	Ref	25.5%	42.9%
ICC	0.029	0.0217	0.017
MOR	1.347	1.293	1.252

OR: Odds Ratio, 95% Cl: 95% confidence interval, T^2^: Area level variance, PCV: Proportional change in Variance, ICC: Intra Class Correlation, MOR: Median Odds Ratio, Model 1: Null model with suburb level random effect, Model2: Model 1 + individual-level factors, Model 3: Model 2+ neighbourhood level IRSD quintiles.

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
