# Peer review of "Examining the Association between Neighbourhood Socioeconomic Disadvantage and Type 2 Diabetes Comorbidity in Serious Mental Illness"

_ijerph, 2019, doi:10.3390/ijerph16203905_

Round 1

Reviewer 1 Report

The current manuscript is a well-written, concise report of your study design and findings. There is some room for improvement by expanding the literature/background to discuss briefly any other studies related to neighborhood effects on either SMI or T2D, even if they did not examine neighborhood effects with comorbidity. This would help expand the discussion of the overall relationship of living environment on health disparities within this context.

The explanation of the index used to calculate neighborhood socioeconomic disadvantage is very useful. In creating the quintiles, does the data reveal that neighborhoods in the higher quintiles for disadvantage are adjacent to each other? Are there any other patterns that emerge when mapping the neighborhoods? In your discussion you mention several factors that could impact policy such as access to health care, fast food restaurants, etc. Is there any data on fresh food markets/food deserts in these areas, as these are prevalent factors for prevention and control of T2D. 

Was data on race/ethnicity collected or utilized to determine if there were any relationships between comorbidity and neighborhood disadvantage? Are there any clusters of groups from certain countries of origin in specific neighborhoods? Were there enough individuals from various country of origin backgrounds to examine this factor or would you conclude that the sample, when broken down into individuals with SMI-T2D comorbidity, is fairly homogeneous?

Author Response

Response to Reviewer 1 comments

1.There is some room for improvement by expanding the literature/background to discuss briefly any other studies related to neighborhood effects on either SMI or T2D, even if they did not examine neighborhood effects with comorbidity. This would help expand the discussion of the overall relationship of living environment on health disparities within this context.

Response: Thank you for this valuable suggestion. We have briefly included in the discussion section , the studies related to neighbourhood disadvantage and SMI or T2D as independent conditions. The following information has been added to the discussion

Our findings are, however, consistent with prior studies,which show significant neighbourhood level socioeconomic inequalities in the distribution of SMI [1-3]and T2D [4-7]as independent conditions. In their systematic review, Mair et al identified 45 studies, of which 37 reported significant associations between neighbourhood characteristics and depression[8]. Similarly, the significant associations between neighbourhood environments and T2D risk was revealed in another systematic review by Dendup et al. [9].

  1. The explanation of the index used to calculate neighborhood socioeconomic disadvantage is very useful. In creating the quintiles, does the data reveal that neighborhoods in the higher quintiles for disadvantage are adjacent to each other? Are there any other patterns that emerge when mapping the neighborhoods?

Response: Thank you for this suggestion. We acknowledge that the information on spatial clustering of neighbourhood disadvantage would strengthen our claim for using it to inform service planning and provisioning. We used Global Moran’s I to investigate spatial autocorrelation or clustering of neighbourhood disadvantage quintiles. Moran’s, I index revealed a significant positive global spatial autocorrelation for neighbourhood disadvantage quintiles (Moran’s I = 0.443673, p = <0.0001) indicating that suburbs with similar neighbourhood disadvantage are clustered geographically. We have also included this information in the methods section as follow.

Global Moran’s I revealed a significant spatial dependence for neighbourhood socio economic disadvantage quintiles (Moran’s I = 0.443673, p = <0.0001)indicating that suburbs with similar relative neighbourhood disadvantage are clustered geographically.

  1. In your discussion you mention several factors that could impact policy such as access to health care, fast food restaurants, etc. Is there any data on fresh food markets/food deserts in these areas, as these are prevalent factors for prevention and control of T2D. 

Response: This study is a part of PhD program of work examining the association between neighbourhoods and T2D comorbidity in SMI. Neighbourhood features that may have an impact on SMI -T2D comorbidity such as neighbourhood fast food environment may be considered in our future studies. However, these data are currently not available for the present investigation.

  1. Was data on race/ethnicity collected or utilized to determine if there were any relationships between comorbidity and neighborhood disadvantage? Are there any clusters of groups from certain countries of origin in specific neighborhoods? Were there enough individuals from various country of origin backgrounds to examine this factor or would you conclude that the sample, when broken down into individuals with SMI-T2D comorbidity, is fairly homogeneous?

Response: We acknowledge this concern raised by reviewer 1. We have utilized the country of birth information in the current study as ethnicity details were not available in our dataset.  Country of birth information were aggregated based on the Standard Australian Classification of Countries produced by the Australian Bureau of Statistics and the descriptive statistics are presented in Table 1. We have now mentioned the non-availability of ethnicity details as a limitation in our discussion section.

We note that individual socioeconomic status, age at diagnosis, ethnicity and number of hospital admissions were not included in this analysis due to the lack of data availability.

References

  1. Dauncey, K., et al., Schizophrenia in Nottingham: Lifelong Residential Mobility of a Cohort. British Journal of Psychiatry, 1993. 163(5): p. 613-619.
  2. Kirkbride, J.B., et al., Social Deprivation, Inequality, and the Neighborhood-Level Incidence of Psychotic Syndromes in East London. 2014. p. 169-180.
  3. Galea, S., et al., Urban Neighborhood Poverty and the Incidence of Depression in a Population-Based Cohort Study. Annals of Epidemiology, 2007. 17: p. 171-179.
  4. Cox, M., et al., Locality deprivation and Type 2 diabetes incidence: A local test of relative inequalities. Social Science & Medicine, 2007. 65: p. 1953-1964.
  5. Cubbin, C., et al., Neighborhood deprivation and cardiovascular disease risk factors: protective and harmful effects. Scandinavian Journal of Public Health, 2006. 34(3): p. 228-237.
  6. Bonney, A.D., et al., Area level socioeconomic disadvantage and diabetes control in the SIMLR Study cohort: Implications for health service planning. 2015: Research Online.
  7. Astell-Burt, T., et al., Understanding geographical inequities in diabetes: Multilevel evidence from 114,755 adults in Sydney, Australia. 2014. p. E68-E73.
  8. Mair, C., A.V.D. Roux, and S. Galea, Are neighbourhood characteristics associated with depressive symptoms? A review of evidence. Journal of Epidemiology and Community Health, 2008. 62(11): p. 940.
  9. Dendup, T., et al., Environmental Risk Factors for Developing Type 2 Diabetes Mellitus: A Systematic Review. International Journal Of Environmental Research And Public Health, 2018. 15(1).

Reviewer 2 Report

The manuscript titled “Examining the association between neighborhood socioeconomic disadvantage and type 2 diabetes comorbidity in serious mental illness” examined the association between neighborhood socioeconomic disadvantage and serious mental illness (SMI) - type 2 diabetes (T2D) comorbidity in an Australian population. Authors draw the hypothesis that neighborhood socioeconomic disadvantage might be positively associated with T2D comorbidity in SMI. Data regarding a large sample of individuals with SMI was retrieved from an institutional platform. Three models of multilevel logistic regression were compared, and results found that residing in the most disadvantaged neighborhoods augmented the likelihood of having SMI-T2D comorbidity of 3.2 times, compared with the most advantaged neighborhoods. Significant geographic variation in the distribution of SMI -T2D also emerged. Authors discuss their results in light of existing literature as well as the limitations of the present study, and gives hints for further research.

I carefully read the manuscript, and I think it may be of interest for the readers of IJERPH. I think there are some minor issues to be addressed. Below there are my comments and suggestions.

The research question of this study is well defined and represent a novelty in the field of epidemiology of mental illnesses in comorbidity with organic conditions, as well as in the investigation of socio-environmental variables which could have a significant impact on the development of comorbidities. The results are appropriately interpreted and significant; the statistics used are pertinent with the aims of the study. The paper is well written and the English-language used is clear. “Materials and Methods” and “Results” sections are well divided and explained. Tables are understandable and well laid out. Discussion follows a commendable logic which elaborates the rationale explained in the Introduction.

I have just one question: why didn’t Authors take into account in statistical modeling other relevant individual level variables, such as age at diagnosis for both conditions or number of hospital admissions for SMI? Usually they are significant factors to be addressed.

Abstract

Page 1 line 23: What do “Q1” and “Q5” means? Even if in the full text they mean “quintiles”, please use full names in the abstract, some readers may not understand. Same goes for “MOR” in the next line.

Minor issues

Please carefully check the manuscript for typos and minor grammar issues.

Author Response

Response to Reviewer 2 comments

  1. why didn’t Authors take into account in statistical modeling other relevant individual level variables, such as age at diagnosis for both conditions or number of hospital admissions for SMI? Usually they are significant factors to be addressed.

Response: Reviewer 2 has raised an important question here. We agree that these factors may be relevant predictors for T2D comorbidity in SMI. However, these details are not available in our dataset. We have included this as a study limitation in our discussion section as below

We note that individual socioeconomic status, age at diagnosis, ethnicity and number of hospital admissions were not included in this analysis due to the lack of data availability.

  1. Abstract:Page 1 line 23: What do “Q1” and “Q5” means? Even if in the full text they mean “quintiles”, please use full names in the abstract, some readers may not understand. Same goes for “MOR” in the next line.

Response: We apologise for this oversight from our part. We have removed the acronyms from the abstract and have replaced them with full terms as below.

Compared with the most advantaged neighbourhoods, residents in the most disadvantaged neighbourhoods had 3.2 times greater odds of having SMI-T2D comorbidity even after controlling for confounding factors (OR 3.20, 95% CI 1.42-7.20). Analysis also revealed significant geographic variation in the distribution of SMI -T2D comorbidity in our sample (Median Odds Ratio = 1.35).

  1. Please carefully check the manuscript for typos and minor grammar issues.

Response: Thank you for pointing this out. We have gone through the manuscript and have checked for typos and grammar mistakes.

Reviewer 3 Report

1, This paper try to investigate the association between neighbourhood socioeconomic disadvantage and SMI-T2D comorbidity. Is there association between neighbourhood socioeconomic disadvantage and T2D-SMI comorbidity?

2, Add a column for the “Individuals with T2D” in table 1.

3, According to table 2 (column 1), only 2.9% of the variance in comorbidity was attributable to between neighbourhood differences. Need more evidence to show that it is a important issue to investigate the association between neighbourhood socioeconomic disadvantage and SMI-T2D comorbidity?

4, It will be interesting to analysis the association between neighbourhood socioeconomic disadvantage and T2D-SMI comorbidity in table 2.

5. More evidence are needed to support the policy implication, that is planning interventions and resourcing public health services based on this study, as only 2.9% of the variance in comorbidity was attributable to between neighbourhood differences.

Author Response

Response to Reviewer 3 comments

  1. This paper try to investigate the association between neighbourhood socioeconomic disadvantage and SMI-T2D comorbidity. Is there association between neighbourhood socioeconomic disadvantage and T2D-SMI comorbidity?

Response: We acknowledge the  potential for a two-way relationship between SMI and T2D. However, this paper is substantively concerned with T2D in persons with SMI and not SMI in persons with T2D. Moreover, we do not have data available to undertake these analyses. We have noted the potential for reverse causation as a study limitation in the discussion section

We also acknowledge the potential for reverse causation as SMI and T2D share a bidirectional association.

  1. Add a column for the “Individuals with T2D” in table 1

Response: Thank you for this suggestion. We only have data on SMI individuals and the SMI individuals with T2D. Individuals with only T2D is not available for investigation and is beyond the scope of this study.

  1. According to table 2 (column 1), only 2.9% of the variance in comorbidity was attributable to between neighbourhood differences. Need more evidence to show that it is a important issue to investigate the association between neighbourhood socioeconomic disadvantage and SMI-T2D comorbidity?

Response: As noted by Diez-Roux, a lower neighbourhood variance value should not preclude important neighbourhood level effects[1]. Misspecification of neighbourhoods, smaller group sizes and even omission of a relevant level 1 variable can all cause under estimation of neighbourhood variance[2]. Low Intra class correlation (ICC) can coexist with important neighbourhood level fixed effects. Several of these examples are available in public health where risk factors explain very little neighbourhood variance but are important predictors of health outcomes[2]. Therefore, small neighbourhood variance does not imply that the effects of neighbourhood variables are not worth investigating. Additionally, Geoffrey Rose had pointed out that even small neighbourhood effects when aggregated at population scales can have a massive impact[3]. This information has also been added to the discussion section

Lower neighbourhood variance in SMI -T2D comorbidity (ICC = 2.9) reported in our study does not preclude important neighbourhood level effects[1]. Misspecification of neighbourhoods, smaller group sizes and even omission of a relevant level 1 variable can all cause under estimation of neighbourhood variance[2]. Low Intra class correlation (ICC) can coexist with important neighbourhood level fixed effects and several of these examples are available in public health where risk factors explain very little neighbourhood variance but are important predictors of health outcomes[2]. Additionally, Geoffrey Rose had pointed out that even small neighbourhood effects when aggregated at population scales can have a massive impact[3].

  1. It will be interesting to analysis the association between neighbourhood socioeconomic disadvantage and T2D-SMI comorbidity in table 2.

Response: This is an interesting question addressed earlier. This paper is concerned with T2D in persons with SMI and investigating SMI in individuals with T2D is beyond the scope of the current study.  Moreover, we do not have data available to undertake these analyses.

  1. More evidence are needed to support the policy implication, that is planning interventions and resourcing public health services based on this study, as only 2.9% of the variance in comorbidity was attributable to between neighbourhood differences.

Response: As discussed earlier, smaller neighbourhood variance does not necessarily imply lack of neighbourhood level effects.  Aggregated at population scale, small neighbourhood effects can have massive impacts as there are so many individuals in a population.  [3]. Moreover, the above study identified significant positive association between neighbourhood socio economic disadvantage and SMI-T2D comorbidity and even the raw rates of SMI-T2D comorbidity were higher in the most disadvantaged neighbourhoods.

References

  1. Diez Roux, A.V., Estimating neighborhood health effects: The challenges of causal inference in a complex world. Social Science and Medicine, 2004. 58(10): p. 1953-1960.
  2. Diez Roux, A.V., Neighborhoods and Health: What Do We Know? What Should We Do? American Journal of Public Health, 2016. 106(3): p. 430.
  3. Rose, G., Sick individuals and sick populations. Int J Epidemiol, 2001. 30(3): p. 427-32; discussion 433-4.

Reviewer 4 Report

The authors of the manuscript Examining the association between neighbourhood  socioeconomic disadvantage and type 2 diabetes comorbidity in serious mental illness have written a good report on their research findings. Overall, the paper is well-written, the methods are correct and specific, and the results are clear. However, I think that the authors need to address some issues before considering it for publication.

There are 3 major issues that the authors need to consider. First, the authors need to address better why the relationship between neighborhood disadvantage and diabetes is especially important in the case of patients with mental illness. It is difficult to understand what the study adds to the literature, and why the relationship between neighborhood disadvantage and diabetes should matter specifically in this case.

Second, I’m worried about the representativeness of Electronic Health Records, as there is not so much information provided.

Third, I’m concerned about selection bias in this study. If you only take people with severe mental illness (which is related with deprivation), those living in high deprivation areas could be more severe cases which are more susceptible to co-morbidities; not because they’ve been exposed to neighborhood factors, but because they are more severe. I’d recommend running some sensitivity analysis using as main outcome (co-morbidities vs NONE of the diseases).

I’d go through some specific comments.

Introduction

L32 Can you be more specific on why DM2 is specifically important for people with serious mental illness?

Methods

L66 Is there any information on the quality of the EHR? How these people are representative of the population?

L67 Can you include more information about the study area? For example, is this area more urban/rural than other Aus areas? How many people live on average in an SSC?

L83 Did you consider exploring how your results differ by the type of mental illness? Or at least different severities.

L94 Do you have any information if deprivation has changed between 2011 and 2017? I’m worried that changes after the economic crisis could bias the relationship

Results

I would suggest the authors follow a recent article on how to improve descriptives tables ( https://www.ncbi.nlm.nih.gov/pubmed/31229583 ); there are some suggestions that I think it will improve the information that the authors provide in table 1.

Discussion

L166 I think that there is a typo and should be 17.3% as reported in results.  

I suggest the authors make the case on why this relationship is important for people with severe mental illness. Also, I suggest the authors to look more in-depth into the mechanisms behind the association; I suggest to look at the recent Bilal et al review on neighborhood factors and diabetes ( https://www.ncbi.nlm.nih.gov/pubmed/29995252 )

Author Response

Reviewer 4 comments

  1. First, the authors need to address better why the relationship between neighborhood disadvantage and diabetes is especially important in the case of patients with mental illness. It is difficult to understand what the study adds to the literature, and why the relationship between neighborhood disadvantage and diabetes should matter specifically in this case.

Response: Thank you for this suggestion. We had briefly mentioned this information in our introduction section. But we have added more information in the revised manuscript to improve the clarity.

Individuals with SMI have 2 to 4 times increased risk of developing type 2 diabetes (T2D) compared with the general population which translates into a reduction of 15 – 20 years in their life expectancies [1-3].A comorbid T2D diagnosis is also associated with other adverse consequences such as increased hospitalisations, greater number of emergency department visits, non-adherence to treatments, higher healthcare utilisation costs, higher risk of cognitive deficit, poor clinical outcomes and decreased quality of life for the mentally ill. [1, 4-10].

Establishing strong evidence of the relationship between neighbourhood disadvantage and SMI –T2D comorbidity is an important step in advancing our understanding of the T2D comorbidity in SMI and the possible associations neighbourhood environments might have with this comorbidity. Moreover, population-based prevention strategies that shift the risk distribution of entire population in a favourable direction are considered more effective and sustainable than the individual based approaches in reducing the disease burden [11]. Understanding these associations may also be useful for health policy makers to develop integrated interventions and to provide greater diversity of care needed to optimally manage the complex needs associated with comorbidity.

  1. I’m worried about the representativeness of Electronic Health Records, as there is not so much information provided.

Response: The public health service data used in this study is clinically coded ,includes up to 50 diagnosis to capture SMI and is highly accurate. The majority of SMI-attributable acute admissions are to public hospitals. Australian National Surveys of Psychosis indicates that 45.6-62.9% of people with SMI reported ≥1 hospital admission for any reason in the previous 12 months [12]. This should have provided us a reasonable coverage given our eight-year data collection period.

  1. I’m concerned about selection bias in this study. If you only take people with severe mental illness (which is related with deprivation), those living in high deprivation areas could be more severe cases which are more susceptible to co-morbidities; not because they’ve been exposed to neighborhood factors, but because they are more severe. I’d recommend running some sensitivity analysis using as main outcome (co-morbidities vs NONE of the diseases).

Response: We acknowledge this concern. We aren’t arguing that neighbourhood disadvantage causes SMI-T2D comorbidity. Individuals with SMI have two to four times  higher risk of T2D comorbidity compared with the general population. We are only seeking to determine if this might be related to contextual factors that exacerbates or buffers against the prevalence of SMI-T2D comorbidity. We also do not have access to data that provides “None’ of the diseases and the suggested analysis is beyond the scope of our study.

  1. Introduction

L32 Can you be more specific on why DM2 is specifically important for people with serious mental illness.

Response: Thank you for this comment. We have modified line 32 as below

Individuals with SMI have 2 to 4 times higher risk of developing T2D compared with the general population which translates into a reduction of 15 – 20 years in their life expectancies. A comorbid T2D diagnosis is also associated with several other adverse consequences for people with SMI such as increased hospitalisations, greater number of emergency department visits, non-adherence to treatments, higher healthcare utilisation costs, higher risk of cognitive deficit, poor clinical outcomes and decreased quality of life.

  1. Methods

L66 Is there any information on the quality of the EHR? How these people are representative of the population

Response: The public health service data used in this study is clinically coded ,includes up to 50 diagnosis to capture SMI and is highly accurate. Majority of acute admissions due to SMI are to public hospitals. Australian National Surveys of Psychosis indicates that 45.6-62.9% of people with SMI reported ≥1 hospital admission for any reason in the previous 12 months [12]. This should have provided us a reasonable coverage given our eight-year data collection period.

  1. L67 Can you include more information about the study area? For example, is this area more urban/rural than other Aus areas? How many people live on average in an SSC?

Response: The study area consists of both urban and rural suburbs. Average population in a suburb is mentioned in the methods section as below

The Illawarra-Shoalhaven region comprised of 167 suburbs with an average land area of 36.56 km2 and 2207 residents each in 2011.

  1. L83 Did you consider exploring how your results differ by the type of mental illness? Or at least different severities

Response: Thankyou for this suggestion. We didn’t have enough individuals in our dataset to undertake analysis by mental illness type without losing the power (Schizophrenia n =110, Bipolar disorder n = 76, Major depression n = 186, other affective disorders n = 13 and other non-affective psychosis n = 76). Information on disease severity were not available for investigation.

  1. L94 Do you have any information if deprivation has changed between 2011 and 2017? I’m worried that changes after the economic crisis could bias the relationship

Response: We acknowledge that this concern is very relevant. However, Australia was largely unaffected by the 2009 Global financial crisis and any difference between 2011 and 2017 would simply reflect the normal economic growth. We did assess the strength of agreement between 2011 and 2016 disadvantage quintiles using weighted kappa analysis. The Kappa statistics revealed a fairly good agreement between the two (k = 0.796) indicating that the deprivation scores have remained relatively similar during these periods. 

  1. I would suggest the authors follow a recent article on how to improve descriptives tables ( https://www.ncbi.nlm.nih.gov/pubmed/31229583 ); there are some suggestions that I think it will improve the information that the authors provide in table 1.

Response: Thank you for this valuable suggestion. We have modified the descriptive table based on the mentioned article. The modified table is as given below.

Table 1. Characteristics of study population Variables

Variables

Individuals with SMI

n= 3816

Individuals with SMI-T2D comorbidity

n = 463

% of persons with SMI who also have comorbidity (95% Cl)

Individual variables

Gender

Female

Male

1848 (48%)

1968 (52%)

245 (53%)

218 (47%)

13.3 (11.8–14.9)

11.1 (9.7–12.5)

Age, years (Mean (SD)

Age, years

18 – 44

45 – 65

65+

43.6 [18.5]

1961 (51%)

1213 (32%)

642 (17%)

58.8 [15.7]

92 (20%)

193 (42%)

178 (38%)

4.7 (03.8–05.7)

15.9 (13.9– 18.0)

27.7 (24.3–31.2)

Country of birth

Australia

Oceania excluding Australia

UK & Ireland

Western Europe

Eastern and central Europe

North East Asia

South East Asia

Central and South Asia

Middle East and North Africa

Sub-Saharan Africa

Americas

3104 (81%)

74 (2%)

212 (6%)

137 (4%)

125 (3%)

17 (0%)

51 (1%)

16 (0%)

39 (1%)

20 (1%)

21 (1%)

339 (73%)

12 (3%)

35 (8%)

29 (6%)

29 (6%)

0 (0%)

6 (1%)

3 (1%)

9 (2%)

0 (0%)

1 (0%)

10.9 (9.9–12.1)

16.2 (9.5–26.2)

16.5 (12.1–22.1)

21.2 (15.2–28.8)

23.2 (16.7–31.3)

0.0 (0–18.4)

11.8 (5.5–23.4)

18.8 (6.6–43.0)

23.1 (12.7–38.3)

0.0 ( 0–16.1)

4.8 (0.9–22.7)

Neighbourhood level variables

IRSD as quintiles

Q1 (Highest)

Q2

Q3

Q4

Q5 (Lowest)

1752 (46 %)

943 (25 %)

620 (16 %)

362 (10 %)

139 (4 %)

229 (49%)

120 (26%)

75 (16%)

34 (7%)

7 (2%)

13.1 (11.6–14.7)

12.7 (10.7–14.9)

12.1 (9.8–14.9)

9.4 (6.8–12.8)

5.1 (2.5–10.0)

IRSD = Index of relative socioeconomic disadvantage

  1. L166 I think that there is a typo and should be 17.3% as reported in results.  

Response: We have corrected this mistake from our side.

References

  1. Holt, R.I. and A.J. Mitchell, Diabetes mellitus and severe mental illness: mechanisms and clinical implications. Nat Rev Endocrinol, 2015. 11(2): p. 79-89.
  2. Ward, M. and B. Druss, The epidemiology of diabetes in psychotic disorders. Lancet Psychiatry, 2015. 2(5): p. 431-451.
  3. DE HERT, M., et al., Physical illness in patients with severe mental disorders. I. Prevalence, impact of medications and disparities in health care. World Psychiatry, 2011. 10(1): p. 52-77.
  4. Wandell, P., et al., Diabetes and psychiatric illness in the total population of Stockholm. J Psychosom Res, 2014. 77(3): p. 169-73.
  5. Ribe, A.R., et al., Long-term mortality of persons with severe mental illness and diabetes: a population-based cohort study in Denmark. Psychol Med, 2014. 44(14): p. 3097-107.
  6. Tirupati, S. and L.E. Chua, Obesity and metabolic syndrome in a psychiatric rehabilitation service. Aust N Z J Psychiatry, 2007. 41(7): p. 606-10.
  7. Šprah, L., et al., Psychiatric readmissions and their association with physical comorbidity: a systematic literature review. BMC Psychiatry, 2017. 17.
  8. Kurdyak, P., et al., Diabetes quality of care and outcomes: Comparison of individuals with and without schizophrenia. General Hospital Psychiatry, 2017. 46: p. 7-13.
  9. Zhang, B.H., et al., Gender differences in cognitive deficits in schizophrenia with and without diabetes. Compr Psychiatry, 2015. 63: p. 1-9.
  10. Han, M., et al., Diabetes and cognitive deficits in chronic schizophrenia: a case-control study. PLoS One, 2013. 8(6): p. e66299.
  11. Rose, G., Sick individuals and sick populations. Int J Epidemiol, 2001. 30(3): p. 427-32; discussion 433-4.
  12. Morgan, V.A., et al., People living with psychotic illness in 2010: The second Australian national survey of psychosis. Australian & New Zealand Journal of Psychiatry, 2012. 46(8): p. 735-752.

Round 2

Reviewer 4 Report

Thank you very much for the thoughtful revision and response. The authors have addressed most of my concerns with the original version, or if they were not able to solve them, they explain why they couldn’t. However, some minor issues should be considered:

I’m still worried about the representativeness of the sample. Maybe only more severe cases are those that are registered in the hospital. Do those 45.6-62.9% of people that report hospital admission go to public hospitals like the ones you get the data for the study? It will help the reader if the information you provide about the study area is given by comparing with Australian standards. For example, is that population density high or low for Australian standards? I’d suggest the authors to include the time difference in the deprivation index as one of the limitations of the study. Despite there is a good correlation between 2011 and 2016; you don’t know if those areas that don’t correlate are somehow related to your exposure or your outcome

Author Response

Response to reviewer comments

1.I’m still worried about the representativeness of the sample. Maybe only more severe cases are those that are registered in the hospital. Do those 45.6-62.9% of people that report hospital admission go to public hospitals like the ones you get the data for the study?

Response: We acknowledge the concern raised here.  We are only claiming a reasonably good data coverage (but not complete coverage) based on the reports from Australian survey of psychosis and our 8-year data collection period. We have acknowledged our data sourcing from only public health hospitals as a limitation in this study.

There are some limitations with our study. First, the cross-sectional study design does not allow us to draw cause-effect conclusions. Second, we used data sourced only from inpatient mental health records and did not consider outpatient and private practice records. However, the Australian National Surveys of Psychosis indicates that 45.6-62.9% of people with SMI reported ≥1 hospital admission for any reason in the previous 12 months [1], which should have provided a reasonable coverage given our eight year data collection period.

  1. It will help the reader if the information you provide about the study area is given by comparing with Australian standards. For example, is that population density high or low for Australian standards?

Response: Thank you for this suggestion. We have tried to include this information now in the methodology section.

The region has a mix of rural and urban influences and is comprised of the local government areas of Kiama, Shellharbour, Shoalhaven and Wollongong. The socio-economic profile of the study area as described by region’s socio economic index scores are comparable to that of NSW and Australian average[2, 3].

  1. I’d suggest the authors to include the time difference in the deprivation index as one of the limitations of the study. Despite there is a good correlation between 2011 and 2016; you don’t know if those areas that don’t correlate are somehow related to your exposure or your outcome

Response: Thank you for this suggestion. The potential for temporal misalignment is now included as a study limitation as below

we acknowledge the potential for temporal misalignment as 2011 relative disadvantage index scores were used in this analysis. Nonetheless a weighted Kappa analysis between 2011 and 2016 disadvantage quintiles revealed a fairly good agreement between the two (k = 0.796) indicating that the deprivation scores have remained relatively similar during these periods.

References

  1. Morgan, V.A., et al., People living with psychotic illness in 2010: The second Australian national survey of psychosis. Australian & New Zealand Journal of Psychiatry, 2012. 46(8): p. 735-752.
  2. Ghosh, A., et al., Using data from patient interactions in primary care for population level chronic disease surveillance: The Sentinel Practices Data Sourcing (SPDS) project. BMC Public Health, 2014. 14(1): p. 557.
  3. Australian Bureau of Statistics., A introduction to socioeconomic indexex of the areas (SEIFA), ABS, Editor. 2011: Canberra.